# Ventilator-Associated Pneumonia in the Neonatal Intensive Care Unit—Incidence and Strategies for Prevention

**DOI:** 10.3390/diagnostics14030240

**Published:** 2024-01-23

**Authors:** Vanya Rangelova, Ani Kevorkyan, Ralitsa Raycheva, Maya Krasteva

**Affiliations:** 1Department of Epidemiology and Disaster Medicine, Faculty of Public Health, Medical University of Plovdiv, 4000 Plovdiv, Bulgaria; ani.kevorkyan@mu-plovdiv.bg; 2Department of Social Medicine and Public Health, Faculty of Public Health, Medical University of Plovdiv, 4000 Plovdiv, Bulgaria; r.raycheva@mu-plovdiv.bg; 3Department of Obstetrics and Gynecology, Neonatology Unit, Faculty of Medicine, Medical University of Plovdiv, 4000 Plovdiv, Bulgaria; maya.krasteva@mu-plovdiv.bg

**Keywords:** ventilator-associated pneumonia, neonate, neonatal intensive care unit, incidence, prevention

## Abstract

The second most prevalent healthcare-associated infection in neonatal intensive care units (NICUs) is ventilator-associated pneumonia (VAP). This review aims to update the knowledge regarding the incidence of neonatal VAP and to summarize possible strategies for prevention. The VAP incidence ranges from 1.4 to 7 episodes per 1000 ventilator days in developed countries and from 16.1 to 89 episodes per 1000 ventilator days in developing countries. This nosocomial infection is linked to higher rates of illness, death, and longer hospital stays, which imposes a substantial financial burden on both the healthcare system and families. Due to the complex nature of the pathophysiology of VAP, various approaches for its prevention in the neonatal intensive care unit have been suggested. There are two main categories of preventative measures: those that attempt to reduce infections in general (such as decontamination and hand hygiene) and those that target VAP in particular (such as VAP care bundles, head of bed elevation, and early extubation). Some of the interventions, including practicing good hand hygiene and feeding regimens, are easy to implement and have a significant impact. One of the measures that seems very promising and encompasses a lot of the preventive measures for VAP are the bundles. Some preventive measures still need to be studied.

## 1. Introduction

The survival rates of preterm neonates have increased in the past decades due to the advances in intensive care treatment. This achievement is the result of the use of antenatal steroids, exogenous surfactant supplementation, improved nutrition, and mechanical ventilation (MV). Unfortunately, mechanical ventilation is associated with a substantial risk of ventilator-associated pneumonia (VAP) [1,2]. This type of nosocomial infection (NI) can be caused by aspirations of secretions, colonization of the respiratory tract, and the contamination of devices and medications used for the treatment of the hospitalized neonate [3].

Ventilator-associated pneumonia is defined as pneumonia not present or incubating at the time of admission and occurring after more than 48 h of MV [4]. This is the second most common healthcare-associated infection (HAI) in neonatal intensive care units (NICUs). VAP is associated with increased morbidity, mortality, increased length of stay in the NICU and hospital costs [5,6,7]. 

Neonates, and more specifically premature babies and those born with a low (<2500 g) or very low birth weight (<1500 g) (LBW and VLBW) are more vulnerable to VAP due to an impaired or immature immune system and the necessity for the use of a combination of invasive devices during their hospital stay. The incidence of VAP in NICUs can vary significantly in different regions from 1 to 63 episodes per 1000 ventilation days which can reflect differences in diagnostic procedures, definitions used, and the burden of disease [8,9,10]. Different strategies have been discussed for the prevention of VAP such as hand hygiene, stress ulcer prophylaxis, head of bed elevation, early extubation, and suction practices [11] but still there is no consensus on which are the most suitable and cost-effective strategies.

This review aims to update the knowledge related to the epidemiology of neonatal VAP, and more specifically, the incidence of this healthcare-associated infection (HAI) and to summarize the possible strategies for prevention.

## 2. Materials and Methods

We performed a narrative review, integrating robust information from studies on the incidence of ventilator-associated pneumonia in NICUs with recommendations for preventing VAP [12]. The initial literature search was conducted in the Medline (PubMed) and Scopus bibliographic databases. The search strategy included the utilization of both MeSH terms and pertinent free-text terms, specifically, (VAP OR ventilator-associated pneumonia) AND (neonate OR neonatal OR infant) AND (incidence OR prevention OR preventative measures). Our search only included articles that were published up until December 2022. 

V.R. and R.R. conducted an initial screening of the titles and abstracts separately. A subsequent evaluation was conducted through a comprehensive examination of the entire text by the same reviewers. The research examined by both reviewers was juxtaposed, and any discrepancies were deliberated over. A study was deemed suitable if it fulfilled the following criteria:(1)The study included patients who were admitted to the neonatal intensive care unit (NICU) and were less than 28 days old at the time of hospitalization;(2)A case–control or cohort study;(3)Ventilator-associated pneumonia (VAP) was suspected 48 h after the initiation of mechanical ventilation;(4)Published in the English language.

Studies were excluded from the review if they met one of the following criteria:(1)It was an animal study;(2)Included participants already diagnosed with VAP;(3)It was a duplicated study;(4)The study was case-report;(5)The time of mechanical ventilation for the included participants was < 48 h or not specified in the study.

## 3. VAP Definition

One of the setbacks when it comes to VAP and comparing data from different studies is the lack of unified criteria and a gold standard for a VAP diagnosis. There are different definitions but there is still no consensus on this matter.

According to the Centers for Disease Control and Prevention (CDC) and National Nosocomial Infections Surveillance (NNIS), VAP is defined as a pneumonia that develops at least 48 h after initiation of mechanical ventilation [4]. To diagnose VAP, the patient needs to exhibit a combination of radiological, clinical, and microbiological findings. One of the drawbacks of the CDC/NNIS definition for VAP for infants younger than 1 year is that there are no specific criteria for newborns and premature infants. Despite this, most of the studies of VAP in NICUs use the CDC/NNIS criteria [13].

There are several other possible definitions for VAP provided by other researchers and study groups. The “Neo-KISS” module of the German National Nosocomial Infection Surveillance System (Krankenhaus Infektions Surveillance System [KISS]) provides a VAP definition for infants with very low birth weight [14]. A Dutch study group established their own definition for VAP in neonates, which are more inclusive than the CDC definitions [7]. The criteria for diagnosis of Neo-KISS module and the Dutch study group are presented in Table 1.

### 3.1. Clinical Findings

For the diagnosis of ventilator-associated pneumonia, the most common clinical criteria that are used are fever, worsening gas exchange, new onset or increasing bradycardia (<80/min) or tachycardia (>200/min), new onset or increasing tachypnea (<60/min) or apnea (>20 s), new onset or increasing dyspnea (retractions, nasal flaring, grunting), and purulent secretions. Unfortunately, those criteria are non-specific. Apisarnthanarak et al. [15] reported that hypothermia (79%) and new-onset tachypnea (63%) were the most common clinical symptoms in extremely preterm neonates. In their study, Khattab et al. [16] conducted a comparison between newborns with and without ventilator-associated pneumonia (VAP). They found that 80% of the neonates with VAP exhibited chest auscultation abnormalities and mucopurulent endotracheal tube discharge.

### 3.2. Radiological Findings

The most common radiologic criteria used to diagnose VAP in neonates are the presence of new or progressive infiltrate, adhesions or fluid in lobar fissures/pleura, or cavitations, pneumatocele, or air bronchograms on chest X-ray. In comparison to infiltrates, the presence of air bronchograms shows a higher sensitivity (58–83%) [17]. In complicated instances, including infants with underlying cardiac or pulmonary disease, sequential chest X-rays (days 0, 2, 3, and 7) aid in the confirmation of VAP. VAP is known to exhibit a wide range of non-specific radiographic abnormalities that can mimic other lung conditions, such as respiratory distress syndrome (RDS) brought on by a surfactant deficiency which can further complicate an accurate diagnosis [18].

### 3.3. Microbiological Findings

Currently, there are two techniques—invasive (bronchoscopic) and minimally invasive (tracheal aspirate)—to obtain a sample for microbiologic examination. Bronchoalveolar lavage (BAL) is the most reliable method for sampling in the neonate population and is highly specific but difficult to employ in all cases and requires experienced specialists due to the small diameter of the endotracheal tube (ETT) [19]. On the other hand, the minimally invasive method for collection of specimens through tracheal aspirates (TA) is easy to use but there is a risk of over-diagnosing VAP, which can lead to the overuse of antibiotics [20]. Baltimore [13] acknowledged the challenges associated with diagnosing ventilator-associated pneumonia (VAP) in the neonatal intensive care unit (NICU) population, namely regarding the interpretation of endotracheal tube (ETT) cultures. The colonization of the endotracheal tube by both Gram-positive and Gram-negative organisms usually happens after 48 h of intubation and MV. This is also the time when the diagnosis of VAP is initially evaluated [21,22]. Hence, distinguishing between colonization and infection is challenging, and the reliability of tracheal aspirate cultures is uncertain.

In order to find out how different professional opinions and criteria can affect the identification of VAP cases, Cordero et al. [23] undertook a study. Using the CDC classification and a positive tracheal aspirate, 37 newborns hospitalized in the NICU were diagnosed with VAP by designated infection control practitioners (ICPs). Of the 37 patients recognized by the ICPs as VAP cases, seven were diagnosed with VAP by the NICU’s neonatologists. Additionally, radiologists were also asked to examine the X-rays of the 37 patients and in 8 of the 11 patients with equivocal signs of infection, they stated that there were radiographic changes suggestive of VAP. These authors came to the conclusion that a single positive tracheal aspirate does not differentiate between airway colonization and VAP, and radiography findings without conclusive clinical and laboratory evidence may be erroneous.

The signs and symptoms that most of the available definitions for VAP use are subjective and this can lead to variabilities in the reported cases between researchers [24,25]. This might be the possible explanation why the CDC introduced the so-called ventilator-associated episode (VAE) definitions to better capture the wide range of complications that might occur during mechanical ventilation [26]. VAEs are brought on by persistent rises in ventilator settings following a period of stable or declining ventilator settings. Pediatric VAE (PedVAE) is defined as an increase in the daily minimum mean airway pressure of 4 cmH_2_O that is sustained for 2 calendar days after 2 days of stable or decreasing daily minimum mean airway pressure, or an increase in the FiO_2_ of 25 points that is sustained for 2 days after 2 days of stable or decreasing daily minimum FiO_2_s in children and neonates [27]. This type of definition and classification is not routinely implemented by NICUs but can help in the future to better differentiate the different complications that might occur in this fragile population during mechanical ventilation.

Despite improvements in other NICU care practices that have considerably increased the survival of extremely low birth weight (ELBW) (<1000 g) infants, the diagnosis and management of VAP in the NICU setting have not evolved over the past few decades. Therefore, there is an urgent need to enhance the diagnostics of VAP in this population of patients through research that emphasizes establishing more sensitive and accurate diagnostic techniques and biomarkers [28].

## 4. VAP Incidence

VAP is a common and severe complication in NICU patients ranging from 1.4 to 7 episodes per 1000 ventilator days up to 16.1 to 89 episodes per 1000 ventilator days in developing countries [29]. VAP is correlated with elevated rates of illness and death, as well as extended periods of hospitalization [15]. These outcomes impose a substantial financial strain on both healthcare systems and patients’ families. We summarized the VAP incidences reported from studies from different regions in Table 2.

From the table, it is evident that there are differences in the incidences of VAP in the different regions which might be due to the economic state of the countries and the definitions used for VAP diagnosis. The differences in the used criteria does not allow us to compare the results and to draw conclusions. A meta-analysis assessing the neonatal healthcare-associated infection in Brazil estimated a pooled VAP incidence density of 7.9 per 1000 ventilator days (95% CI 1.1–55.5) [54]. The National Healthcare Safety Network reported lower pooled mean VAP rates in children’s hospital NICUs than in general hospital NICUs among neonates weighing 750 g or less (1.68 vs. 2.68; *p* = 0.002) and neonates weighing 751–1000 g (1.10 vs. 2.31; *p* = 0.02) [55]. The International Nosocomial Infection Control Consortium (INICC) reported data from 50 countries with a pooled mean VAP rate of 9.02 episodes per 1000 ventilator days in 2016 [56], whereas the last report of the Consortium collecting data from 45 countries reported a pooled mean VAP rate of 6.9 episodes per 1000 ventilator days [57].

## 5. Prevention Strategies

VAP as a complication in infants is multifactorial and the measures for prevention of this HAI should also entail multiple interventions or different steps in the care of newborns which operate synergistically. Within the range of treatments, certain measures, such as practicing proper hand hygiene and implementing appropriate feeding practices, are both straightforward and highly beneficial. However, there are other interventions that require additional research in order to be properly assessed. The preventive strategies can be categorized as general measures and specific measures (Figure 1).

### 5.1. General Measures for Prevention of VAP

#### 5.1.1. Hand Hygiene

Hand hygiene is one of the most important practices to reduce HAIs [3,58]. Research has shown that adhering to hand hygiene guidelines effectively decreases the likelihood of cross-contamination from VAP organisms found in the gastrointestinal tract. These pathogens can be transmitted through the hands of medical personnel, causing colonization in the respiratory and digestive systems [58]. Healthcare providers have the potential to introduce pathogens into the mouth of a patient through activities such as oral hygiene, bathing, diaper changing, tracheal suctioning, enteral feeding, and handling medical devices. The World Health Organization (WHO) released an advisory outlining the recommended practice of using alcohol-based antiseptics for hand hygiene in hospital environments [59]. WHO has also specified in the guidelines the 5 Moments for Hand Hygiene that hygienic hand disinfection is required (1) before examining/touching a patient, (2) after contact with a patient, (3) before clean/aseptic procedures, (4) after body fluid exposure, and (5) after touching objects in the patient’s surroundings.

In the NICU, hand hygiene compliance has led to reduction in respiratory tract infections from 3.35 to 1.06 infections per 1000 patient-days in one study and decreased VAP rates by 38% in another study [60,61]. Regarding the improvement in handwashing compliance, it has been reported that among nurses, the compliance is higher compared to physicians [62]. A quality improvement initiative in Ireland focusing on improving hand hygiene compliance reported a decrease in VAP rates from 9.8 episodes per 1000 ventilator days in the pre-intervention phase to 6.1 episodes per 1000 ventilator days in the post-intervention phase [61].

#### 5.1.2. Decontamination

One of the possible ways for the transmission of pathogens in the hospitals are different objects that might be used for the treatment of neonates. Some microorganisms have the ability to survive for extended periods in the environment and contaminate surfaces in closer proximity to the patient. Mattresses and incubators both offer warm, moist environments that are conducive to pathogen growth and have been linked to outbreaks caused by *Klebsiella* spp. [63]. Based on a recent meta-analysis of 4165 newborns and 108,035 hospital days, preterm infants who are treated in single rooms as opposed to open bay units have a decreased risk of developing sepsis [64]. Environmental contamination in hospitals has been reported to be lower with employing improved cleaning and disinfection techniques utilizing aerosolized hydrogen peroxide, ultraviolet C, and pulsed xenon ultraviolet radiation systems than with traditional techniques [65].

#### 5.1.3. Antibiotic Stewardship

Numerous investigations have demonstrated that recurrent exposure to antibiotics induces the colonization of pathogenic, commonly drug-resistant bacteria. To minimize the likelihood of drug-resistant bacteria and fungal colonization, evidence-based guidelines to prevent unnecessary antibiotic use have been proposed [66]. The rational use of antibiotics in NICUs is one of the practices which is often part of the care bundles for the prevention of VAP [67].

#### 5.1.4. Feeding

Critically ill patients in NICUs can experience gastroesophageal reflux caused by low or nonexistent pressure in the lower esophageal sphincter. This condition can be brought on by a variety of causes, including drugs, opiates used for sedation, low blood pressure, sepsis [68], or when the neonate was born prematurely. Another strategy that has been suggested to contribute to the development of VAP is the aspiration of microorganisms from stomach secretions [58]. Strategies that can decrease neonatal gastric reflux including avoiding excessive stretching of the abdomen, adjusting the duration and timing of feeding, and considering the positioning of the body [5,69,70]. Regarding the impact on VAP, Kusahara et al. [71] reported that there is no general agreement on the optimal way to administer enteral feeding to infants and children.

### 5.2. General Preventive Strategies Which Need Further Evaluation

#### 5.2.1. Probiotics

The prolonged administration of antibiotics, postponing the initiation of enteral feeding, and nursing in incubators are all associated with the depletion of beneficial gut bacteria such as *Bifidobacterium* and *Lactobacilli* spp. This leads to the proliferation of harmful microorganisms and abnormal colonization of the gut. Probiotic supplementation appears to be a good technique for improving the composition of the intestinal microbiota, which could also be used to prevent neonatal VAP and sepsis [72]. However, the two largest trials, ProPrems [73] and PiPS [74], revealed no change in the incidence of late-onset sepsis (LOS) in preterm infants who received probiotics compared to those who did not. There are still uncertainties regarding the efficacy of probiotics and the best bacterial strains and dose to utilize, despite the fact that over 50 trials involving more than 11,000 patients have been carried out [75].

#### 5.2.2. Lactoferin

Lactoferin is an iron-binding glycoprotein which can be found in human milk. It limits the quantity of free iron that pathogenic bacteria can access while fostering the development of commensal bacteria. A trial with human bovine lactoferin [76] showed promise in reducing the rate of infections in premature newborns but another randomized trial of supplementation with bovine lactoferin did not report a decrease in the risk of LOS [77].

### 5.3. Specific Measures for Prevention of VAP

#### 5.3.1. Bundles

The concept of care bundles is not new. These measures consist of three to five evidence-based practices which, implemented together, can lead to much greater improvement in the care for the patients and can contribute to a decrease in the incidence of HAIs [78]. The Institute for Healthcare Improvements (IHI) introduced the adult ventilator bundle in December 2004. It consists of five evidence-based practices: administering daily oral hygiene with chlorhexidine, elevating the head of the patient’s bed at an angle of 30° to 45°, conducting daily assessments to determine if the patient is ready for extubation, and providing prophylaxis for peptic ulcer illness or against deep vein thrombosis [79]. Research has shown that implementing individual therapies can effectively decrease adult VAP rates. When these interventions are combined, the rates can be reduced to zero [80].

The bundles for preventing VAP in NICUs aim to minimize the need for mechanical ventilation and ET intubation wherever possible, as well as to lessen the amount of organisms that can be transferred to the lungs and colonized there. There is a lack of robust studies among the neonatal and pediatric population supporting the use of standardized bundle of care practices for the prevention of VAP. However, there are reports of care practices which, implemented together, have led to a decrease in the rates of VAP [81,82].

It is important to note that care bundles cannot replace infection control but can be additional tools in the continuous improvement of the services provided to patients. Care bundles result from a knowledge of the procedures and psychology required to effect change. They offer the framework for the realization and application of evidence-based practices. Some of the studies reporting results from the implementation of bundles to reduce the rates of VAP are presented in Table 3.

A quality improvement initiative performed in a Chinese NICU [86] reported a significant reduction in the VAP rates (48.84 VAP episodes in the pre-intervention period and 20.86 VAP episodes in the post-intervention period) after the introduction of bundled measures. The measures included reinforcement of hand hygiene practices, rational waste disposal, enhancement of patient isolation and ventilator disinfection, periodic educational activities on VAP prevention, daily assessments of the need for MV, and the rational use of antibiotics.

In 2008, Brilli et al. [83], in a study on neonates, implemented a bundled approach for the prevention of VAP and reported a reduction from 7.8 to 0.5 episodes per 1000 ventilator days and savings of approximately USD 2.4 million (9).

Ceballos et al. [81] implemented a nurse-driven quality intervention aimed at reducing the incidence of central line-associated bloodstream infections (CLABSIs) and ventilator-associated pneumonia. The authors reported 31% reduction in ventilator days and estimated cost-savings of USD 300,000.

A study by Rosenthal et al. [85], carried out in 15 NICUs in 10 developing countries, reported a 33% reduction in VAP rates after implementing a bundled approach. Additionally, during the intervention, the authors reported increased hand hygiene compliance by the NICU staff (62% in baseline period vs. 81% in intervention period).

#### 5.3.2. Endotracheal Intubation, Suction Practices and Early Extubation

During endotracheal intubation, there is a risk of aspiration of pathogens from the oropharynx and the stomach [71]. The risk of VAP is even higher if the neonate requires an immediate intubation or if the endotracheal tube needs to be reinserted within 72 h after extubation [71]. To minimize the chance of introducing harmful microorganisms into the lower respiratory tract, healthcare workers should ensure that the ET tube is sterile before intubation and use an entirely new tube for each attempt. In order to avoid unintentional removal of the endotracheal tube, it is necessary to ensure that there are sufficient staff members present [91].

One of the major risk factors for VAP is the duration of MV [37,42,49]. Oral secretions accumulate in the posterior pharynx of the intubated patient as a result of their impaired swallowing. In order to minimize micro-aspiration of secretions, it is necessary to perform suctioning of the mouth and back of the throat, along with oral hygiene, before any procedures that involve movement of the endotracheal (ET) tube, such as moving the patient, manipulating the ET tube, suctioning, or retaping [92]. The 2003 CDC guidelines outline the benefits of employing a closed system rather than an open system for ET suction, noting that if the catheter can be used indefinitely without changing, it may reduce environmental contamination and reduce expenses [58]. In research by Cordero et al. [93] comparing closed versus open suction systems in NICUs, VAP rates and mortality were comparable; nevertheless, most nurses stated that the closed system was simpler, quicker, and more tolerated by patients.

In the past, instillation of saline water was used to remove ET secretions during suction. The routine use of saline is not recommended due to the physiological and psychological effects for the patient [94]. Additionally, contamination of the vial containing the saline water can occur when twisting off the plastic cap.

#### 5.3.3. Head Positioning

The IHI adult ventilator bundle recommends that the head of bed (HOB) be maintained at 30–45 degrees to decrease aspiration of secretions [79]. For the neonatology patients, the recommendation is for a 15 to 30° elevation although a 15° inclination is the limit for many NICU incubators. It has been demonstrated that preserving at least a 15° head elevation in neonates during breathing is associated with a lower incidence of micro-aspiration of stomach contents [95]. Dutta et al. proposed a 30° HOB elevation and the use of a left lateral side position in VLBW neonates with gastroesophageal reflux [70].

Maintaining the neonate during mechanical ventilation in a midline and lateral position may facilitate the accumulation of oral secretions in the lateral buccal mucosa, thereby reducing the likelihood of collecting in the subglottic region surrounding the uncuffed ET tube. According to Aly et al. [96], the incidence of tracheal colonization in the neonatal population due to the contamination of the oropharynx can be decreased from 87% when in a supine posture to 30% when in a lateral position.

#### 5.3.4. Ventilator Circuit Changes

The ventilator circuit (VC) is commonly contaminated with microorganisms, and this contamination is considered a significant risk factor for the occurrence of VAP [97]. Therefore, it is logical to assume that more frequent ventilator circuit replacements could reduce VAP rates. Contrarily, the most frequent source of VAP is thought to be endogenous, and the patients’ endogenous commensals are the microorganisms that colonize the ventilator circuits. Therefore, it is also thought that the colonization of the ventilator circuit has little effect on the eventual development of VAP, provided that rigorous aseptic procedures are observed when handling and disassembling the circuits to avoid exogenous contamination with pathogenic bacteria.

The CDC recommends changes of the VC only when the equipment is visibly soiled or malfunctioning [98]. A prospective study in a PICU in Thailand demonstrated that the extension of the interval between VC changes from 3 to 7 days did not increase the occurrence rate of VAP and was associated with a significant cost reduction [99]. A recent meta-analysis assessing the frequency of ventilator circuit changes effect on preventing VAP reported that there is no evidence to suggest that ventilator circuits can be safely left unchanged until visibly soiled in neonates and children, although such a practice can lead to a decrease in hospital costs [100].

#### 5.3.5. Educational Interventions

Numerous studies have demonstrated that educational intervention programs can significantly lower VAP rates [101,102]. The burden of VAP in the NICU can be reduced with sufficient training and application of VAP prevention techniques, as nurses are frequently responsible for the continuous care of ventilated newborns in NICUs for 24 h at a time [102]. A study conducted in Thailand found a substantial decrease in the crude mortality rate and VAP rates following the adoption of instructional programs (40.5% vs. 24%, *p* < 0.001, and 12.3% vs. 8.7%, *p* < 0.001, respectively) [103]. The interventions aiming to increase the knowledge of the staff regarding HAIs, including VAP, are often part of the care bundles and increasing the awareness towards this important topic through such interventions can help decrease the rates of HAIs.

#### 5.3.6. Limitations

We should outline some limitations that were found throughout the review process. The requirement for English-language publications may have impacted the composition of the final article sample and the completeness of the references. Furthermore, the exclusion of grey literature and the reliance solely on PubMed and Scopus as data sources, without the inclusion of unpublished publications, may have restricted the breadth of the search. Consequently, the significance of certain data elements may have been undervalued or overvalued due to the potential bias in the interpretation of the findings of the included studies. Further examination of optimal methodologies and resolutions to the acknowledged limitations may be of relevance for a prospective study.

## 6. Conclusions

VAP continues to be a significant complication in NICUs. The variations in the prevalence of VAP across different regions may be attributed to both the economic conditions of the countries and the criteria employed for diagnosing VAP. The pathogenesis of VAP is multifactorial and the measures for the prevention of this HAI should also entail multiple interventions or different steps in the care of newborns which operate synergistically. One of the measures that seems very promising and encompasses a lot of the preventive measures for VAP are the bundles. Some preventive measures still need to be studied.

## Figures and Tables

**Figure 1 diagnostics-14-00240-f001:**
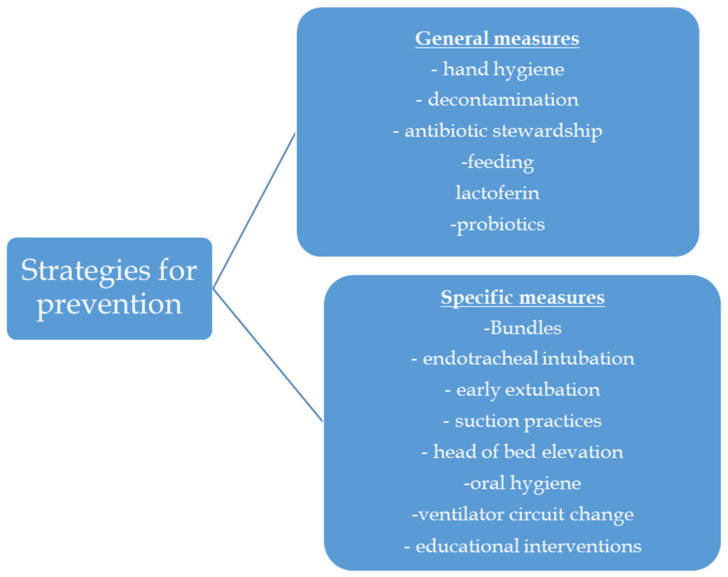
Strategies for prevention of VAP.

**Table 1 diagnostics-14-00240-t001:** Definitions for ventilator-associated pneumonia.

Neo-KISS Module [14]	Van der Zwett et al. [7]
Radiological findings New or progressive infiltrate Consolidation Pleural or interlobar effusion	Clinical findings One of the following: Purulent sputum Changes in sputum characteristics Deterioration of ventilation conditions
AND Deterioration of gas exchange, drop in saturation	Radiological findings New emergence or progression of the following Infiltration Consolidation Pleural adhesion Pleural effusion
AND FOUR of the following criteria New/increased bradycardia (<80/min) or New increased tachycardia (>200/min) New/increased tachypnea (>60 min) or New/increased apnea (>20/s) Purulent tracheal secretions Increased respiratory secretions Isolation of a microorganism from tracheal secretion New/increased dyspnea (retractions, nostril flaring) Temperature instability/fever/hypothermia CRP > 2.0 mg/dL I/T ratio > 0.2	Microbiological findings * Isolation of a pathogenic microorganism or detection of a bacteria/viral antigen in tracheal aspirate, bronchial secretion, or sputum

* If no microorganisms have been isolated in order to diagnose pneumonia, administrations of antimicrobial therapy for at least seven days prescribed by a physician should be present.

**Table 2 diagnostics-14-00240-t002:** Reported incidences of VAP in NICUs.

Study	Study Design	Diagnostic Criteria	Incidence	Most Common Pathogen	Risk Factors
African (AFR) and Eastern Mediterranean Region (EMR)					
Gohr et al. [30]	Prospective observational cohort 140 neonates	Radiographic Clinical Laboratory Microbiologic	27.2%	*Klebsiella* spp. *S. aureus* *Candida* spp.	Reintubation Use of sedatives
Khattab et al. [16]	Prospective observational 85 neonates	Radiographic Clinical Microbiological	55.2%	*S. aureus* *Klebsiella* spp. *Candida* spp.	Low birth weight MV
Afjeh et al. [31]	Prospective cohort81 neonates	Radiographic Clinical	11.6 episodes	*E. coli* *K. pneumoniae*	Purulent sputum
Badr et al. [14]	Prospective observational 56 neonates	Radiographic Clinical Laboratory Microbiologic	57.1%	*Klebsiella* spp. *S. aureus* *Pseudomonas* spp. *E. coli*	Low birth weight Duration of MV Prematurity
ELMeneza et al. [32]	Prospective observational 91 neonates	Radiographic Clinical Microbiological	43.2 episodes/1000 ventilator days	*Klebsiella* spp. *Pseudomonas* *Staph aureus*	Prematurity RDS Reintubation Duration of MV
Region of the Americas (AMR)					
Dudeck et al. [33]	Prospective observational 137 NICUs	CDC/NNIS definition	1.0 episodes/1000 ventilator days for neonates < 750 g 0.1 episodes/1000 ventilator days for neonates > 2500 g	Not reported	Not reported
Patrick et al. [34]	Prospective cohort 173 NICUs	CDC/NNIS definition	1.6 episodes/1000 ventilator days (2007) 0.6 episodes/1000 ventilator days (2012)	Not reported	VLBW
Urzedo et al. [35]	Prospective cohort 4615 neonates	Radiographic Clinical Microbiological	3.2 episodes/1000 ventilator days	*Coagulase (-) Staphylococcus*	Not reported
Romanelli et al. [36]	Prospective observational 886 neonates	CDC/NNIS definition	5.7 episodes/1000 ventilator days	*Staphylococcus aureus* *Klebsiella* spp. *Enterobacter cloacae*	Not reported
South East Asian Region (EASR) and Western Pacific Region (WPR)					
Mir ZH et al. [37]	Prospective observational 96 neonates	Radiographic Clinical microbiological	68.96 episodes/1000 ventilator days	*Klebsiella* spp. *E. coli* *Acinetobacter*	Birth weight <1500 g Duration of MV
Vijayakanthi et al. [38]	Retrospective observational 265 neonates	Radiographic Clinical Microbiological	22.2 episodes/1000 ventilator days	*Klebsiella* spp.	Repeated intubations Unstable initial Cardiopulmonary assessment
Ibrahim et al. [39]	Descriptive correlational 1090 neonates	CDC/NNIS definition	5.7 episodes/1000 ventilator days	*S. aureus* *Klebsiella* spp. *Acinetobacter* spp.	Birth weight Gestational age
Katoch et al. [40]	Prospective observational 37 neonates	CDC/NNIS definition	23.3%	Not reported	Preterm birth (<37 g.w.)
Kawanishi et al. [41]	Retrospective observational 71 neonates ≤2000 g	RadiographicClinical MicrobiologicalFoglia criteria	8.44 episodes/1000 ventilator days	*S. aureus*, *P. aeruginosa*	BW < 626 g
Lee et al. [42]	Retrospective observational 605 neonates	RadiographicClinical Microbiological	7.1 episodes/1000 ventilator days	*K. pneumoniae* *B. cepacia*	Longer duration of intubation TPN
Navoa et al. [43]	Prospective observational 1813 neonates	RadiographicClinical Microbiological	9.6 episodes/1000 ventilator days	*Acinetobacter* spp. *Pseudomonas* spp. *Enterobacter* spp.	Not reported
Thatrimontrichai et al. [44]	Prospective cohort 128 neonates	CDC/NNIS definition	10.1 episodes/1000 ventilator days	*Acinetobacter baumannii* *Stenotrophomonas maltophilia*	BW ≤ 750 gsedative medication use
Deng et al. [45]	Retrospective case–control 349 patients	Foglia definition	25.6 episodes/1000 ventilator days	*Klebsiella* spp. *Acinetobacter baumannii*	Birth weight reintubation Respiratory symptoms
Cai et al. [46]	Prospective observational 1159 neonates	CDC/NNIS definition	48.8 episodes/1000 ventilator days	*Acinetobacter baumannii* *Klebsiella pneumoniae* *Coagulase negative staphylococcus*	Not reported
European region (EUR)					
Demirbag et al. [47]	Point-prevalence 47 neonates	CDC/NNIS definition	38.2%	*Klebsiella* spp. *MRSA*	Not reported
Dell Orto et al. [48]	Prospective, population-based cohort 199 neonates	CDC/NNIS definition	17.1 episodes/1000 ventilator days	*E. coli* *Klebsiella* spp. *Staph haem.*	Not reported
Wojkowska et al. [49]	Prospective cohort 1695 neonates	NEO-KISS definition	18.2 episodes/1000 ventilator days	*CONS* *P. aeruginosa* *A. baumannii*	Duration of MV
Cernada et al. [19]	Prospective observational cohort	CDC/NNIS definition	10.9 episodes/1000 ventilator days	*P. aeruginosa* *S. aureus* *Polymicrobial* (16.7%)	Duration of MV
Scamardo et al. [50]	Prospective observational 1265 neonates	CDC/NNIS definition	20%	*P. aeruginosa* (28%), *Stenotrophomonas maltophilia* (20%) *CONS* (20%)	Not reported
Geslain et al. [51]	Prospective observational 381 neonates	CDC/NNIS definition	8.8 episodes/1000 ventilator days	*Enterobacter cloacae* *Staph.* spp. *Klebsiella* spp.	BW < 1000 g Higher SNAPP score
Leistner et al. [52]	Patient-based prospective 33 048 VLBW neonates	NEO-KISS definition	2.3 episodes/1000 ventilator days	*Staph aureus* *CONS* *Klebsiella* spp.	Not reported
Yalaz et al. [53]	Prospective cohort 600 neonates	CDC/NNIS definition	13.76 episodes/1000 ventilator days	*Stenotrophomonas maltophilia* *Kl. pneumoniae* *Pseudomonas* *aeruginosa*	Not reported

**Table 3 diagnostics-14-00240-t003:** Bundles for prevention of VAP.

Study	Interventions Included in the Bundle	VAP Rates	Mortality Rates
Brilli et al. [83] 2008	Head of bed elevation, daily assessment of readiness for extubation while providing oral care, administering medication to prevent peptic ulcers, practicing proper hand hygiene, changing ventilator circuit if visibly soiled or malfunctioning	Pre-intervention 7.8 episodes/1000 ventilator days	Not reported
		Post-intervention 0.5 episodes/1000 ventilator days	
Pepin et al. [84] 2012	Proper hand hygiene, meticulous intubation technique, assessment of readiness for extubation, thorough disinfection of the environment and equipment, effective management of bedside patient care routines	Pre-intervention 8.5 episodes/1000 ventilator days	Not reported
		Post-intervention 2.5 episodes/1000 ventilator days	
Rosenthal et al. [85] 2012	Hand hygiene, daily assessment of readiness for extubation, oral care with an antiseptic solution, use of noninvasive ventilation when possible, change in ventilator circuit only when visibly soiled	Pre-intervention 17.8 episodes/1000 ventilator days	Not reported
		Post-intervention 12.0 episodes/1000 ventilator days	
Zhou et al. [86] 2013	Hand hygiene, assessment of readiness for extubation, closed endotracheal suctioning, educational activities, weekly changing of the ventilator circuit, rational use of antibiotics	Pre-intervention 48.8 episodes/1000 ventilator days	Pre-intervention 14.0%
		Post-intervention 20.8 episodes/1000 ventilator days	Post-intervention 2.7%
Azab et al. [87] 2015	Head of bed elevation, daily assessment of readiness for extubation, oral care, peptic ulcer prophylaxis, hand hygiene, changing ventilator circuit if visibly soiled or malfunctioning	Pre-intervention 36.4 episodes/1000 ventilator days	Pre-intervention 25.8 %
		Post-intervention 23.0 episodes/1000 ventilator days	Post-intervention 17.3%
Gocke et al. [88] 2018	Adherence to hand hygiene guidelines, readiness to wean assessment, ventilator circuit evaluation and changing the circuit only when visibly soiled or malfunctioning, periodic draining and discarding of ventilator circuit condensate, bed head elevation to 10–13 degrees, oral care	Pre-intervention 7.3 episodes/1000 ventilator days	Not reported
		Post-intervention 2.7 episodes/1000 ventilator days	
Jahan et al. [89] 2018	Hand hygiene, daily assessment of extubation readiness, use of non-invasive ventilation when possible, head of bed elevation, oral care, changing ventilator circuit when visibly soiled	Pre-intervention 59%	Pre-intervention 68.2%
		Pre-intervention 26.3%	Post-intervention 52.6%
Pinilla-González et al. [90] 2021	Healthcare training, hand hygiene, sterile management of airways, avoiding reintubation, oral care, head of bed elevation, changing ventilator circuit only when visibly soiled, tube feeding for 60–120 min	Pre-intervention 11.8 episodes/1000 ventilator days	Pre-intervention 21.3%
		Post-intervention 1.9 episodes/1000 ventilator days	Post-intervention 13.2%

## Data Availability

Not applicable.

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
