# Peer review of "Ventilator-Associated Pneumonia in the Neonatal Intensive Care Unit—Incidence and Strategies for Prevention"

_diagnostics, 2024, doi:10.3390/diagnostics14030240_

Round 1
Reviewer 1 Report
Comments and Suggestions for Authors
The review article entitled " Ventilator-associated pneumonia in the neonatal intensive care unit – incidence and strategies for prevention " is well written and has a good Framework for ventilator-associated pneumonia in neonatal intensive care. This article provides useful information. However, there are some comments that need to be revised as follows:
1. Line 12: “neonatology” should be “neonatal”
2. Line 22: the “VAP car bundles” should be “VAP care bundles”
3. Line 59-60: “newborn intensive care units” should be “NICUs”
4. Line 81-82: The authors have introduced a VAP (Ventilator-Associated Pneumonia) definition specifically for infants with very low birth weight based on the [KISS] study. However, it is imperative for the authors to elucidate the precise details of this VAP definition as utilized in [KISS]. Additionally, providing an explanation of the VAP definition employed by the Dutch group would be equally essential.
5. Line 108, 120: The authors employed the term "tracheal aspirate" on line 108 and subsequently used the term "ETT cultures" on line 120. It is important to clarify whether these two terms are synonymous.
6. Line 261: If "NI" refers to "NICU", which stands for Neonatal Intensive Care Unit, it should be clarified. Additionally, I recommend that the authors avoid using "NI" as an abbreviation for nosocomial infection, as it may cause confusion since "NI" can be interpreted as referring to the same meaning as "NICU".
7. Line 266: the authors should discuss “prophylaxis against deep vein thrombosis” in preventing VAP.
8. Line 275: the “the rate o VAP” should be “the rate of VAP”
9. Line 284: The citation of “a Chinese NICU[84]” should be [86], not [84].
10. Line 323: “In the past instillation of saline water was used to remove….”, please add a comma after in the past.
11. Line 330: “….recommendation is for 15 to 30 ° elevation Although a 15 ° inclination is the limit for many”, I thought “Although” should be “although”.
Author Response
We thank the reviewer for this very important and insightful feedback to improve the paper. We have comprehensively addressed all the suggested comments on areas that can be improved. Thank you.
- Line 12: “neonatology” should be “neonatal”
Thank you for the comment. We have corrected it.
- Line 22: the “VAP car bundles” should be “VAP carebundles”
It is corrected in the revised version.
- Line 59-60: “newborn intensive care units” should be “NICUs”
Thank you for mentioning, we have done corrections.
- Line 81-82: The authors have introduced a VAP (Ventilator-Associated Pneumonia) definition specifically for infants with very low birth weight based on the [KISS] study. However, it is imperative for the authors to elucidate the precise details of this VAP definition as utilized in [KISS]. Additionally, providing an explanation of the VAP definition employed by the Dutch group would be equally essential.
Thank you for the comment. We have added a new Table (Table 1) in the revised version of the manuscript with details of both definitions and the criteria used to diagnose VAP.
- Line 108, 120: The authors employed the term "tracheal aspirate" on line 108 and subsequently used the term "ETT cultures" on line 120. It is important to clarify whether these two terms are synonymous.
Thank you for the comment ETT cultures in the text refer to tracheal aspirate cultures and we have made corrections in the text for better understanding.
- Line 261: If "NI" refers to "NICU", which stands for Neonatal Intensive Care Unit, it should be clarified. Additionally, I recommend that the authors avoid using "NI" as an abbreviation for nosocomial infection, as it may cause confusion since "NI" can be interpreted as referring to the same meaning as "NICU".
Thank you for the comment. We have made corrections and substituted NI in the text with HAI so that it is not confusing to the reader.
- Line 266: the authors should discuss “prophylaxis against deep vein thrombosis” in preventing VAP.
We have not discussed deep vein thrombosis prophylaxis in neonates as part of a bundle approach as this has not been included in the studies assessing bundle interventions, and this prevention measure is still debatable and not accepted.
- Line 275: the “the rate o VAP” should be “the rate ofVAP”
Thank you for the comment. We have corrected it.
- Line 284: The citation of “a Chinese NICU [84]” should be [86], not [84].
We have made the changes required.
- Line 323: “In the past instillation of saline water was used to remove….”, please add a comma after in the past.
Thank you we have corrected it.
- Line 330: “….recommendation is for 15 to 30 ° elevation Although a 15 ° inclination is the limit for many”, I thought “Although” should be “although”.
This has been addressed and correction was made.
Reviewer 2 Report
Comments and Suggestions for Authors
In this paper, authors present an overview of ventilator associated pneumonia in neonatal intensive care. The article is well written and clear.
However I had some struggle to understand the methodology you used.
Thus:
1. Line 58: you mention you performed a "modified version of a narrative review", without explaining what you mean by this. Please make clear what kind of methods you used in the study.
2. I was expecting to find outa number of papers you find at your initial search, what kind of literature did you find and how you selected papers you have taken into consideration for the study.
3. Also, it will be very useful to indicate and justify inclusion and exclusion criteria.
4. In this kind of study, usually readers expect to see what were the important limitations in your methodology. Please mention about them.
5. I noticed your list of References is not prepared in accordance with the MDPI guidelines, please rearrange this section.
Author Response
We thank the reviewer for this very important and insightful feedback to improve the paper. We have comprehensively addressed all the suggested comments on areas that can be improved. Thank you.
Following are the answers to the comments raised.
Points 1-3
- Line 58: you mention you performed a "modified version of a narrative review", without explaining what you mean by this. Please make clear what kind of methods you used in the study.
- I was expecting to find outa number of papers you find at your initial search, what kind of literature did you find and how you selected papers you have taken into consideration for the study.
- Also, it will be very useful to indicate and justify inclusion and exclusion criteria.
In our review we have used the following checklist (https://legacyfileshare.elsevier.com/promis_misc/ANDJ%20Narrative%20Review%20Checklist.pdf ) and we have referred to one article that has been used as a guide for conducting the review. In the revised version of the manuscript, we have added additional information on the steps we have taken in the review process. As in a narrative review, it is not imperative to include the number of papers included; we have not provided this information.
In the revised version of the manuscript, we have made corrections in the Materials and Methods section, and more specifically, we have added the following clarifications:
V.R. and R.R. conducted an initial screening of titles and abstracts separately. A subsequent evaluation was conducted through a comprehensive examination of the entire text by the same reviewers. The research examined by both reviewers was juxtaposed, and any discrepancies were deliberated over. A study was deemed suitable if it fulfilled the following criteria:
1) The study included patients who were admitted to the Neonatal Intensive Care Unit (NICU) and were less than 28 days old at the time of hospitalization
2) A case-control or cohort study
3) Ventilator-associated pneumonia (VAP) was suspected 48 hours after the initiation of mechanical ventilation.
4) Published in the English language
Studies were excluded from the review if they met one of the following criteria:
1) Animal studies
2) Included participants already diagnosed with VAP
3) Duplicated study
4) The study was case-report
5) The time of mechanical ventilation for the included participants was < 48 h or not specified in the study
4.In this kind of study, usually readers expect to see what were the important limitations in your methodology. Please mention about them.
Thank you for the suggestion. We have added a Limitations section with the following text:
We should outline some limitations that were found throughout the review process. The requirement for English-language publications may have impacted the composition of the final article sample and the completeness of the references. Furthermore, the exclusion of grey literature and the reliance solely on PubMed and Scopus as data sources, without the inclusion of unpublished publications, may have restricted the breadth of the search. Consequently, the significance of certain data elements may have been undervalued or overvalued due to the potential bias in the interpretation of the findings of the included studies. Further examination of optimal methodologies and resolutions to acknowledged limitations may be of relevance for a prospective study.
- I noticed your list of References is not prepared in accordance with the MDPI guidelines, please rearrange this section.
Thank you for the comment. We have revised the reference list in accordance with the guidelines.
Round 2
Reviewer 2 Report
Comments and Suggestions for Authors
The paper was improved by taking into consideration reviewers suggestions.
Author Response
Dear Reviewer,
Thank you very much for the positive feedback on the current status of the manuscript and your valuable comments, which we do believe made our paper better.